# Geomorphology of a Holocene Hurricane Deposit Eroded from Rhyolite Sea Cliffs on Ensenada Almeja (Baja California Sur, Mexico)

**Markes E. Johnson** [1,*] **, Rigoberto Guardado-France** [2] **, Erlend M. Johnson** [3] **and Jorge Ledesma-Vázquez** [2]

1   Geosciences Department, Williams College, Williamstown, MA 01267, USA
2   Facultad de Ciencias Marinas, Universidad Autónoma de Baja California,
    Ensenada 22800, Baja California, Mexico; rigoberto@uabc.edu.mx (R.G.-F.); ledesma@uabc.edu.mx (J.L.-V.)
3   Anthropology Department, Tulane University, New Orleans, LA 70018, USA; erlend.johnson@gmail.com
*   Correspondence: mjohnson@williams.edu; Tel.: +1-413-597-2329

**Abstract:** This work advances research on the role of hurricanes in degrading the rocky coastline within Mexico's Gulf of California, most commonly formed by widespread igneous rocks. Under evaluation is a distinct coastal boulder bed (CBB) derived from banded rhyolite with boulders arrayed in a partial-ring configuration against one side of the headland on Ensenada Almeja (Clam Bay) north of Loreto. Preconditions related to the thickness of rhyolite flows and vertical fissures that intersect the flows at right angles along with the specific gravity of banded rhyolite delimit the size, shape and weight of boulders in the Almeja CBB. Mathematical formulae are applied to calculate the wave height generated by storm surge impacting the headland. The average weight of the 25 largest boulders from a transect nearest the bedrock source amounts to 1200 kg but only 30% of the sample is estimated to exceed a full metric ton in weight. The wave height calculated to move those boulders is close to 8 m. Additional localities with CBBs composed of layered rock types such as basalt and andesite are proposed for future studies within the Gulf of California. Comparisons with selected CBBs in other parts of the world are made.

**Keywords:** coastal boulder deposit; hurricane storm surge; hydrodynamic equations; Gulf of California (Mexico)

## 1. Introduction

Hurricane Odile was one of the most destructive storms to strike the Mexican state of Baja California Sur in terms of infrastructure damage [1]. It made landfall just after midnight on September 14, 2014 at Cabo San Lucas on the southern tip of the Baja California peninsula as a Category 4 hurricane packing sustained wind speeds of 144 km/h. Tracking into the Gulf of California, the maximum wind speed fell to 113 km/h by the time the storm reached the town of Loreto located 375 km to the northeast later the same day. As it advanced from under a foot-print diameter of 600 km, the system's counter-clockwise rotation spun out storm bands with the strongest winds and wind-driven waves generated from its energetic right-front quadrant. Quite aside from damage to public and private property of concern to civil authorities, erosion due to coastal flooding and the direct impact of wave activity against natural shorelines is a separate issue of interest to physical geographers and marine geologists.

Outwash from uplands through flooded stream beds has the capacity not only to transport terrestrial sediments to the coast but also to reconfigure unconsolidated shore deposits such as beaches

and estuary tidal bars. Moreover, rocky shorelines are subject to incremental erosion from repeated storms and long-shore currents over time. A previous contribution from our team [2] focused on Holocene events during the last 10,000 years related to the physical erosion of rocky shores on Isla del Carmen, one of the larger fault-block islands in the gulf with 95% coverage by rocky shores. The laterally coherent coastal boulder bed (CBB) that resides 12-m above mean sea level on the east side of Isla del Carmen is distinct due its source from limestone strata vulnerable to storm waves on the outer lip of a marine terrace. Limestone accounts for only a small part of the rocky coast around that particular island, which is dominated by igneous rocks. Based on a coastal survey by Backus et al. (2009) using satellite imagery [3], igneous rocks are represented by granodiorite, andesite, basalt and other volcanic sediments to account for 34% of the shoreline in the western Gulf of California (including islands). By comparison, limestone amounts to only 7.5% of the whole.

An eye-witness account filmed during Hurricane Odile from a landmark home built into limestone cliffs north of Loreto at Ensenada Basilio recorded waves that crashed over coastal prominences at a height of 8 m above mean sea level [4]; see also Supplementary Materials. Horizontal rain reached the inner-most part of the residential compound set back from the cliff edge by some 45 m. At nearby Ensenada Almeja (Clam Bay), a north-oriented headland is formed by igneous rocks with a prominence falling from a high of 18 to 6 m above mean sea level at its distal tip. Theoretically, those cliffs are vulnerable to erosion from wave shock arriving from the east, during which winds from a tropical depression would cross from one side of the headland to the other. The object of this study is the asymmetrical boulder bed forming a semi-ring deposit exclusively on one side of the headland within Ensenada Almeja that partially restricts the bay's opening. This is the first analysis of its kind dealing with igneous rocks that form CBBs in the Gulf of California. Overall, a wide range of energy sources capable of rocky-shore erosion and CBB development include the daily tides, seasonal wind patterns that influence long-shore currents, episodic storms, and tsunamis. A secondary goal is to provide information on additional CBBs throughout the Gulf of California formed by igneous boulders. Common patterns in the physical geography of such features suggest a novel approach forward in the study of CBBs within an active zone of subtropical storms impacting continental margins. Useful comparisons also are made with notable CBBs elsewhere in the world.

## 2. Geographical and Geological Setting

The Gulf of California is a marginal sea between the Mexican mainland and the Baja California peninsula with a NW-SE axis 1100 km long and a 180-km wide opening to the Pacific Ocean at its southern end (Figure 1a). The sea's mean annual sea-surface temperature (SST) is 24° which is higher than the norm of 18° in the adjacent ocean and the mean average rainfall on the peninsula amounts to only 15.3 cm [5]. During relatively infrequent impact by hurricanes, conditions are dramatically altered especially in terms of heavy rainfall over desert terrain lacking thick plant cover. Tropical storms that immerge in the East Pacific Ocean typically form off the coast of Acapulco below 15° N latitude and track northward, turning outward to the west before reaching the southern tip of the Baja California peninsula at 23° N latitude. Storms that stray across the Baja California peninsula into the Gulf of California are called *chubascos*, only some of which amount to disturbances of hurricane strength. Other influences that contribute to the aggradation of coastal sediments include daily tides and seasonal winds that intermittently funnel down the axis of the gulf from the northwest with an average azimuth from N to S from December to March. This is shown by a tightly constrained rose diagram (Backus and Johnson, 2009, their figure 10.1 D), based on the orientation of structures in 84 coastal sand dunes throughout the region [6]. Strong winds capable of generating sea swells given sufficient fetch are known to blow persistently for days at a time, with gusts between 8 and 10 m/s not uncommon [7,8].

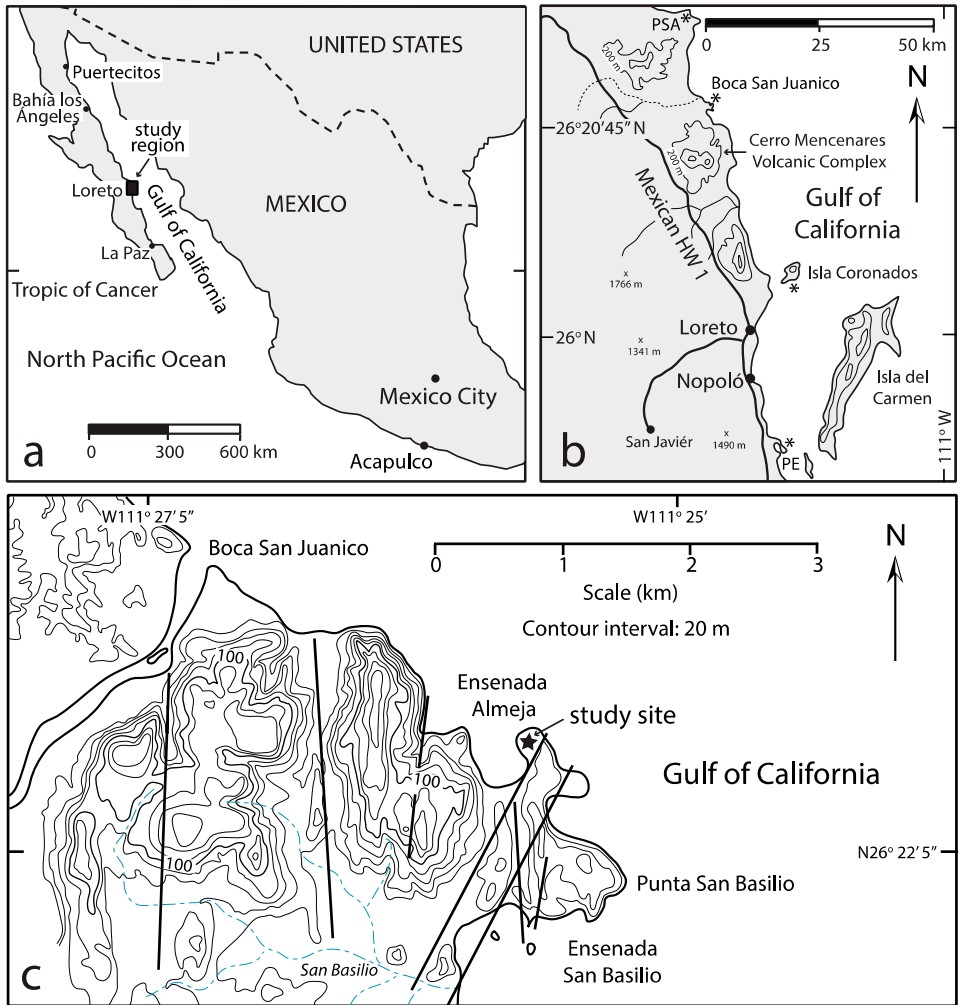

**Figure 1.** Locality maps showing Mexico's Baja California peninsula and Gulf of California; (**a**) Mexico and border area with the United Sates, denoting key villages or cities with dots and the study region marked by a square; (**b**) Region around Loreto in Baja California Sur, marking coastal boulder beds (*) at Puerto Escondido (PS), Isla Coronados, Ensenada Almeja near the Boca San Junico and Punta San Antonio (PSA); (**c**) Study site at Ensenada Almeja.

The geology of the San Basilio area 30 km north of Loreto (Figure 1b) incorporates the study site around Ensenada Almeja (Figure 1c), which features a landscape dominated by igneous rocks including banded rhyolite, hyaloclastite, massive rhyolite and volcanic ash [9]. Fossil-bearing limestone beds deposited around and above massive rhyolite domes are consistent with an assignment to the Zanclean Stage within the Lower Pliocene. Pleistocene deposits of consolidated dune limestone are seated above an eroded rhyolite shelf on the east side of Ensenada Almeja and extend inland through a north-south valley connected to Ensenada San Basilio. Mapping of fault lines throughout the region suggests that the low ground occupied by the Almeja CBB is separated from an adjacent rhyolite dome by a normal fault (Figure 2) parallel to a well-defined fault crossing through part of the headland to the east with a trend of N28°E to S28°W [8].

## 3. Materials and Methods

### 3.1. Data Collection

Ensenada Almeja was visited on 28 and 29 April 2019, when the original data for this study were collected based on a sample of 100 boulders divided equally among four transects crossing the CBB.

A Brunton compass and meter tape were used to lay out the transects, three of which conform to a N-S or E-W axis (Figure 2). The other transect was laid out parallel to the cliff line near the boundary fault. Various conventions exist for the differentiation of sedimentary clasts but the definition for a boulder followed in this exercise is that proposed by Wentworth (1922) for a clast equal to or greater than 256 mm in diameter [10]. No upper limit for this category is found in the geological literature, although Ruban et al. (2019) championed the term "megaclast" for boulders of extraordinary size [11]. That term is an appropriate descriptor for some of the boulders in the Almeja CBB.

The largest 25 boulders along each transect with centers spaced from 1 to 1.5 m apart were measured manually along three principle axes (long, intermediate and short). Triangular plots were employed to show variations in boulder shape, following the design of Sneed and Folk (1958) for river pebbles [12]. Data on the maximum and minimum lengths perpendicular to each other from individual boulders were fitted to bar graphs to show size variations from one transect to the next.

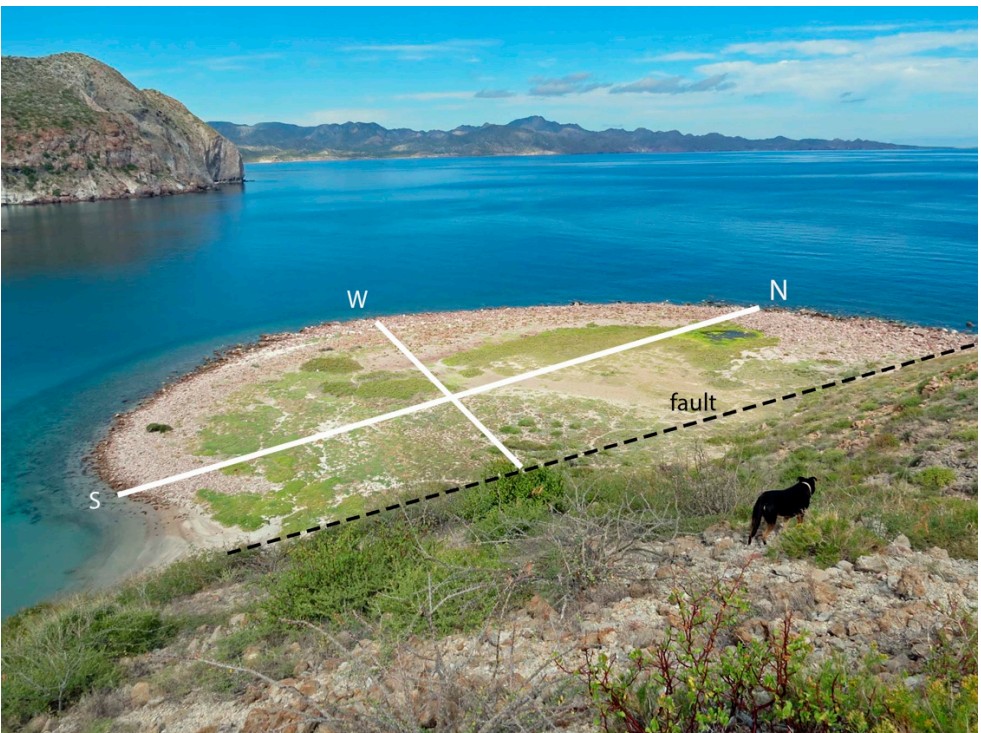

**Figure 2.** Northwest view across Ensenada Almeja coastal boulder bed (CBB) from an elevation of 16 m on the adjacent fault-bound headland formed by rhyolite bedrock (dog for scale on ridge).

A representative cobble of banded-rhyolite was collected from the Almeja CBB for laboratory treatment at Williams College, where it was weighed, and its volume determined as a function of equal displacement when submerged in a tank of water. Prior to immersion, the porous rhyolite sample was water-proofed by spraying it with Thompson's Water Seal [TM] (The Thompson's Co., Cleveland, OH, USA).

A DJI Phantom-2 drone [TM] (DJI, Nanshan District, Shenzhen, China) was flown over the Almeja headland and CBB to provide an overview of the study area and key reference points for construction of a detailed site map.

### 3.2. Hydraulic Model

After the weight and density of a banded-rhyolite sample is determined in the laboratory, a hydraulic model may be applied to predict the energy needed to shift larger rhyolite blocks from the headland outcrop and deposit them as a CBB in Ensenada Almeja. Along with shape, size and density, the pre-transport environment of coastal boulders factors into the wave height required for

detachment and removal. Boulders derived from a weathered surface with deep joints at right angles are influenced mainly by lift force, alone. This requires somewhat higher waves to initiate transport compared to boulders already siting in a submerged position. To initiate motion of a loosened block, the lift force must overcome the force of restraint minus buoyancy, provided the block has separated completely from the basement substrate. Herein, the general formula used to calculate wave height related to CBB development is taken from the work of Nott [13], used for estimation of storm waves.

$$H_s \geq \frac{(P_s - P_w/P_w)^{2a}}{C_d(ac/b^2) + C_1}$$

where $H_s$ = height of the storm wave at breaking point; $u = (gH)^{0.5}$ and $\partial = 1$; $a, b, c$ = long, intermediate and short axes of the boulder (m) $P_s$ = density of the boulder (tons/m$^3$ or g/cm$^3$), $C_d$ = drag coefficient, $C_m$ = coefficient of mass (= 2) and $C_1$ = lift coefficient (= 0.178);

$$u = \text{instantaneous flow acceleration } (= 1 \text{ m/s}^2)$$

A variation on this formula applied exclusively to joint-bounded boulders is as follows [13]:

$$H_s \geq (P_s - P_w/P_w) \, a/C_1$$

## 4. Results

### 4.1. Sample Density Calculation

The banded-rhyolite sample retrieved for laboratory analysis measures $15 \times 8 \times 5$ cm on three axes perpendicular to one another. Due to irregularities in shape, however, it is not accurate to equate volume with a simple multiplication of the measurements in cubic centimeters (600 cm$^3$). The weight of the sample was found to be 843.5 g. After treatment making the sample water-tight, submergence in water registered a displacement equal to 390 mL. Dividing mass by volume yielded a density of 2.16 for banded rhyolite. Checking the laboratory result for volume against the mathematical result, it was found that the actual volume is only 65% of the latter. All of the boulders in the Almeja CBB are crudely shaped with dimensions similar to a shoe box but with irregularities. Roughly the same adjustment regarding irregular shapes was taken into account when correcting for the estimated boulder weight based on the three-dimensions measured in the field.

The banded rhyolite and hyaloclastite typical of the San Basilio region are seldom found elsewhere on the Baja California peninsula, although smaller but similar rhyolite domes occur on Isla San Luis [14] to the north. The style of Pliocene volcanism at San Basilio and Quaternary volcanism at Isla San Luis are favorably compared with Quaternary rhyolites from islands in the Tyrrhenian Sea off Italy [15,16]. Although the text-book value for the specific gravity of massive rhyolite is commonly given as 2.5, values ranging between 1.6 and 2.8 were calculated for samples by Calanchi et al. (1993), from the Aeolian islands [16]. Our result for the banded rhyolite from Ensenada Almeja falls midway within that range.

### 4.2. Aerial Photography

An aerial photo from the drone flight on 28 April 2019 (Figure 3) shows the distinct partial-ring shape of the Almeja CBB appended against the flank of basement rocks on the west side of the headland. The boulder field does not extend into the ring's central depression. Width of the enclosing ring is greatest adjacent to the front of the headland and most narrow some 120 m back along the fault margin with basement rocks. A small patch of standing water occurs within the circular green zone close to the north inner wall of the ring.

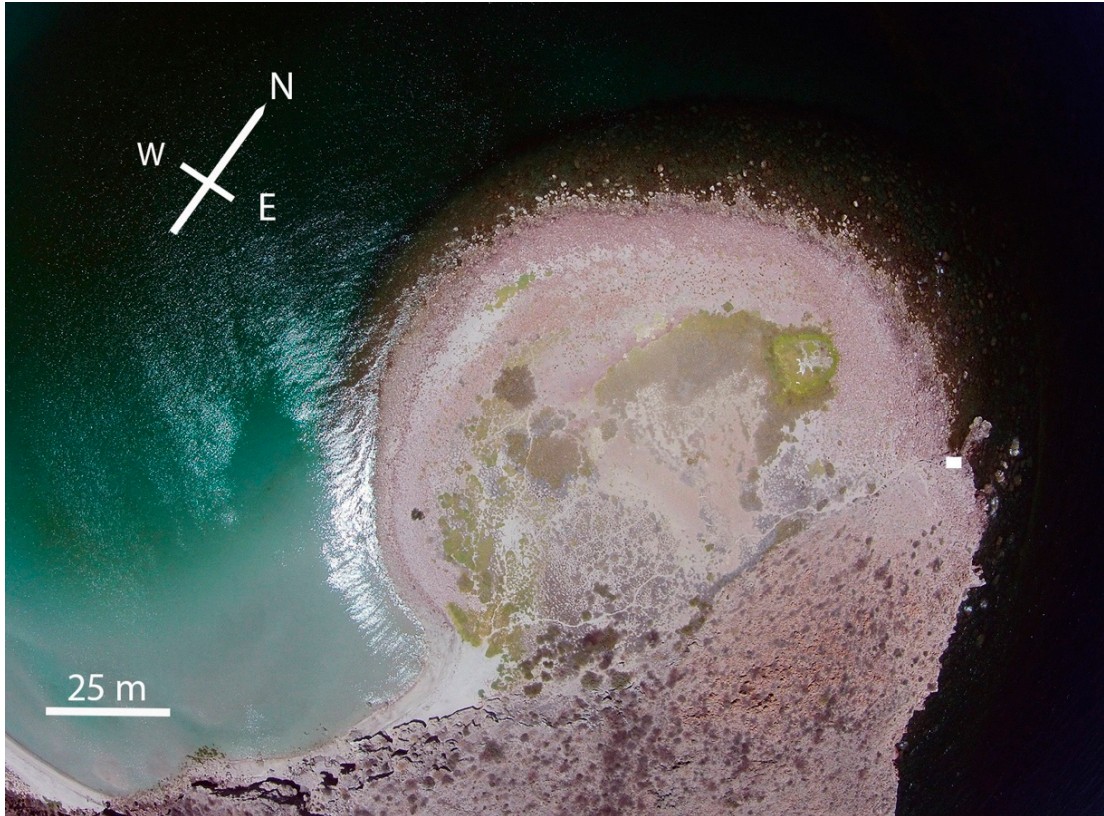

**Figure 3.** Aerial view directly above the Ensenada CBB showing the half-ring arrangement of the deposit on the west flank of the bedrock ridge. White rectangle on the nose of the headland marks location shown in the following field photograph (Figure 4).

## *4.3. Source Rock and Natural Weathering*

Banded rhyolite is the dominant lithology on the headland forming the east flank of Ensenada Almeja. It is the exclusive source of eroded boulders in the Almeja CBB. The natural state of weathered bedrock is exposed in sea cliffs near the tip of the headland (Figure 3, white rectangle). From a ground view near sea level (Figure 4), the bedrock is found to be dissected by bedding planes and vertical fissures intersecting at right angles that outline oblong shapes similar to common shoe boxes. On average, such blocks are roughly three quarters of a meter in length, a half meter wide and a third of a meter in height. The apparent source of mechanical retreat worked on the rhyolite sea cliffs is hydraulic pressure exerted by wave surge against joints in the bedrock. Loosened blocks on bedding planes are poised to slide down slope into the sea aided by the force of gravity.

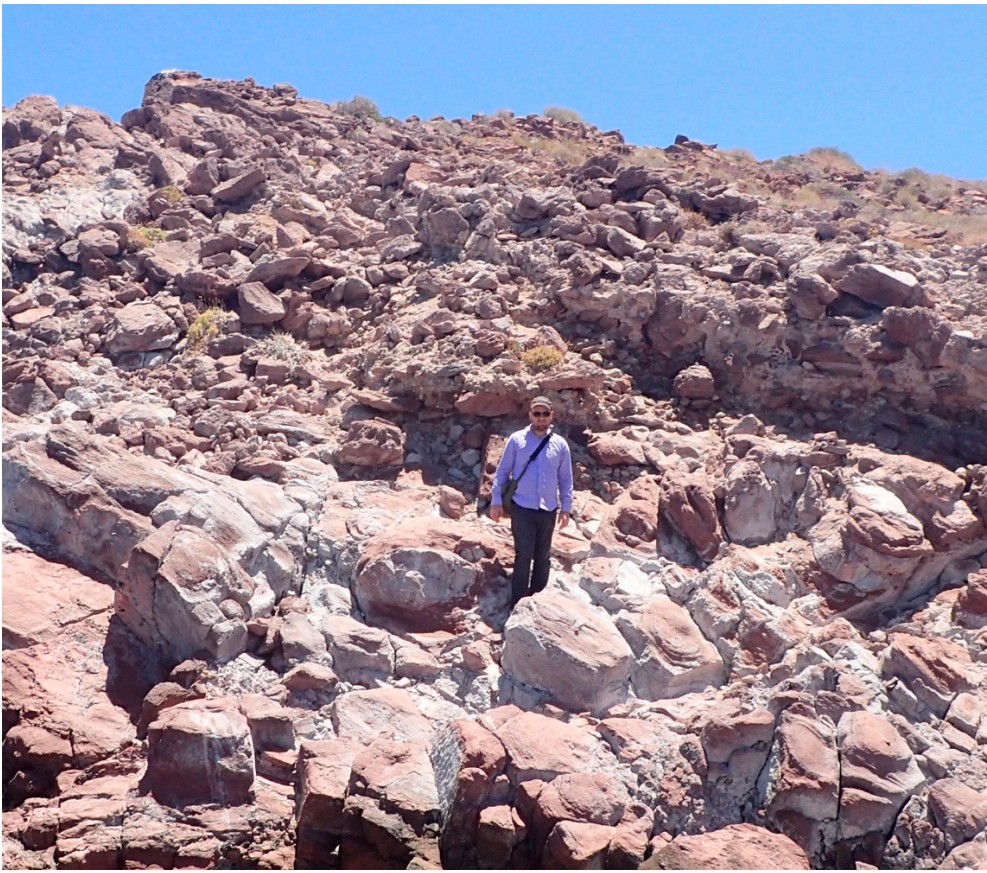

**Figure 4.** Bedrock exposure of banded rhyolite on the outer headland of Ensenada Almeja (see Figure 3 for location), showing pattern of intersecting vertical joints and bedding planes that facilitate boulder production under wave attack.

### 4.4. Mapping and Installation of Sampling Transcects

With the addition of ground measurements, the aerial photo (Figure 3) was consulted to draw a map of the Almeja CBB (Figure 5) from which different parts of the deposit could be quantified. The structure's subaerial exposure amounts to a total area of 13,000 m$^2$, of which the boulder field around the rim occupies close to half at 6500 m$^2$. As of 28 April 2019, swampy ground and open water occupied approximately 200 m$^2$ adjacent to the inner north wall of the ring. Bare ground covers a larger area and it is likely that over-wash of sea water enlarges the area of wet ground from time to time. The location and orientation of four transects across the CBB are marked on the map, the longest of which extends for 50 m sub-parallel to the bedrock escarpment in the northern part of the structure. The shortest is the N-S oriented transect crossing the south rim of the structure. In the central part of the Gulf of California, the maximum tidal range varies by as much as 2.75 m [17]. The subtidal portion of the Almeja CBB likely adds another 2500 m$^2$ to the boulder field for a total of 9000 m$^2$.

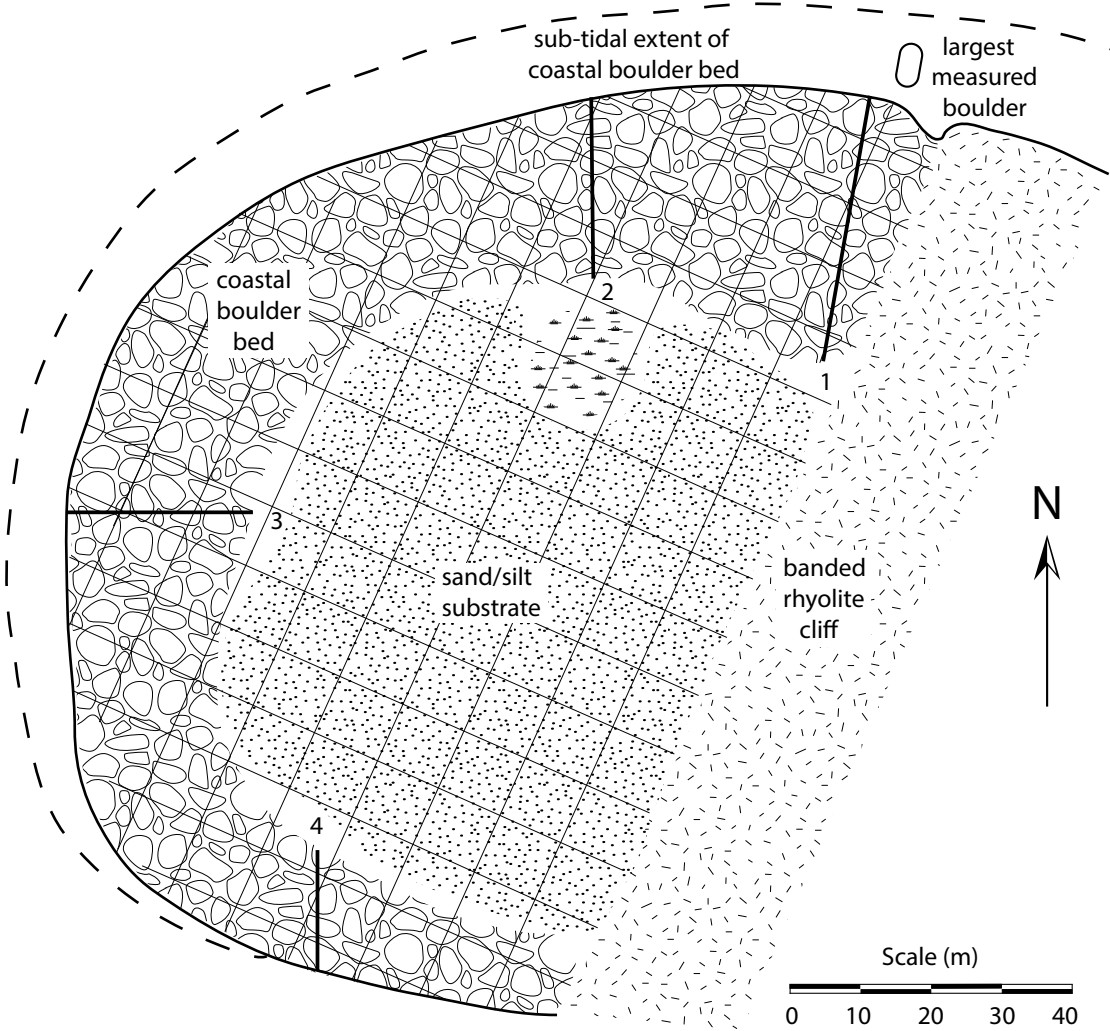

**Figure 5.** Map of the Ensenada Almeja CBB with location of four transects used to collect data on boulder size and shape. Each square on the superimposed grid represents 100 square meters.

The tape measure laid out along transect 1 is shown in place (Figure 6a), with a mega-boulder adjacent to the first author. It turned out to be the second largest individual boulder measured in the Almeja CBB, with a long axis of 268 cm and an estimated volume of 2000 cubic decimeters. The weight of the boulder is estimated conservatively to be on the order of 3450 kg. The height of the boulder ring above the swampy ground on transect 2 stands at 1.85 m (Figure 6b). At its farthest extent seaward to the north, the base of large boulders sits in water up to 2 m below mean sea level. The largest rhyolite boulder encountered during offshore exploration from the northeast corner of the Almeja CBB is estimated to measure 6 m in length by 3 m wide and 2 m high (Figure 5).

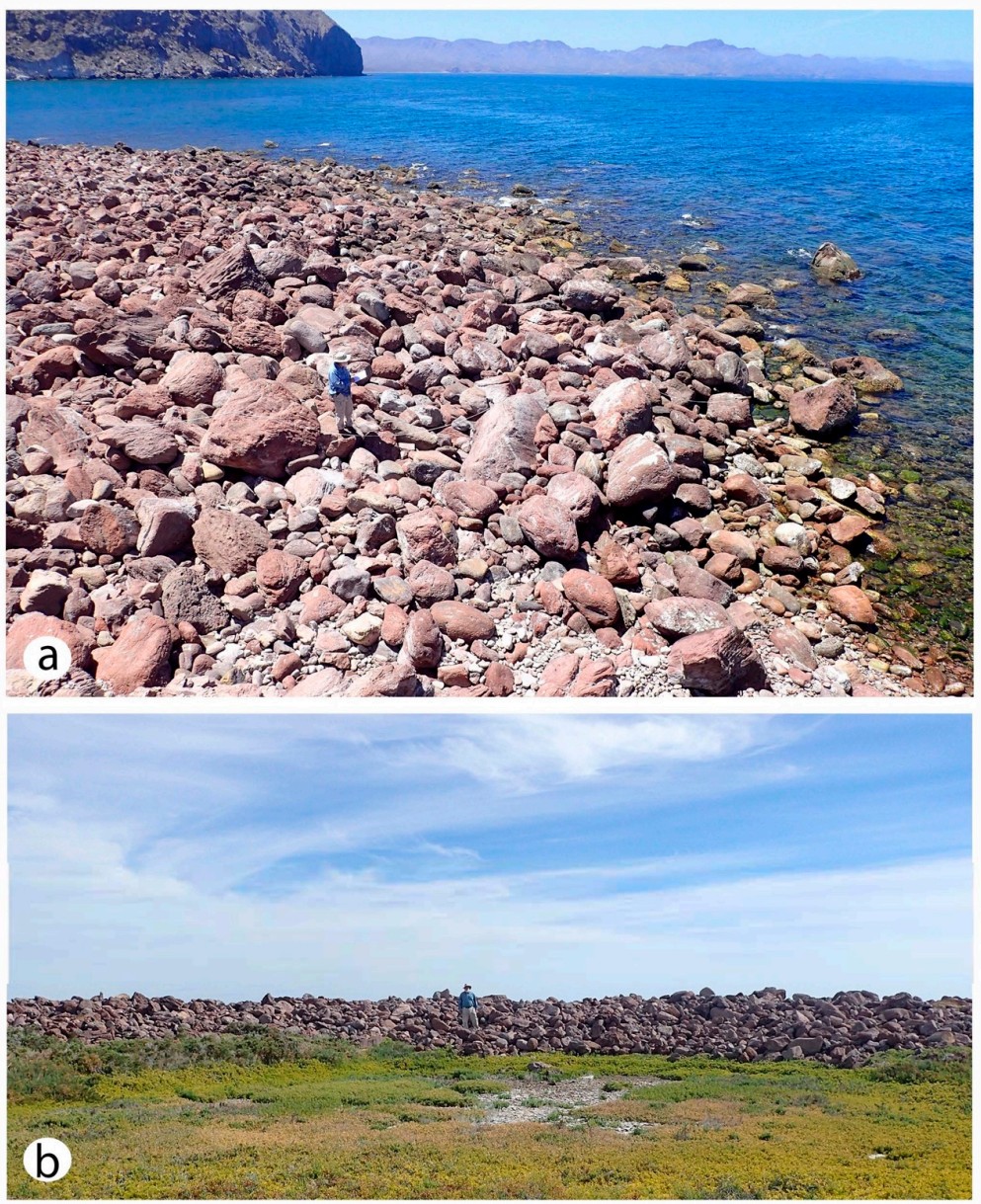

**Figure 6.** Northern part of the Almeja CBB; (**a**) View to the northwest across Ensenada Almeja crossing the path of Transect 1 (see Figure 5)—the largest boulders in the deposit belong to this sector (figure for scale); (**b**) View due north, showing the deposit's inner wall diagonal to Transect 2 (figure for scale).

*4.5. Analysis of Boulder Shapes*

Raw data on boulder size in three dimensions collected from each of four transects across the Almeja CBB are available in Appendix A (Tables A1–A4). Points representing individual blocks grouped by transect are plotted on a set of Sneed-Folk triangular diagrams (Figure 7a–d), showing the actual variation in shapes.

Those points clustered closest to the center of the diagram are most faithful to an intermediate value. With only rare occurrences registered, the absence of points at the top of the triangle signifies that no boulders eroded from equidimensional cubes are present in the sample. Also, the lack of points in the lower, left tier of the triangle demonstrates that squarely plate-shaped blocks are completely absent from the assemblies. Overall, the points grouped from different transects trace similar trends in direction from the center to the lower right tier of the diagram. The recurrent relationship denotes the presence of subpopulations of elongated boulders in the shape of shoe boxes. The significance of

such diagrams puts a heavy emphasis on the thickness of parent rhyolite flows and the spacing of intersecting vertical joints in the bedrock (Figure 4). This result has a direct bearing on the relative ease with which individual blocks might be pried loose from the bedrock by wave action and fall into the sea. The process of rounding is expected to occur from the grinding of blocks against one another under wave surge against the headland. Although the trends in shape are similar among the samples from different transects (Figure 7), the plots have no bearing on variations in boulder size.

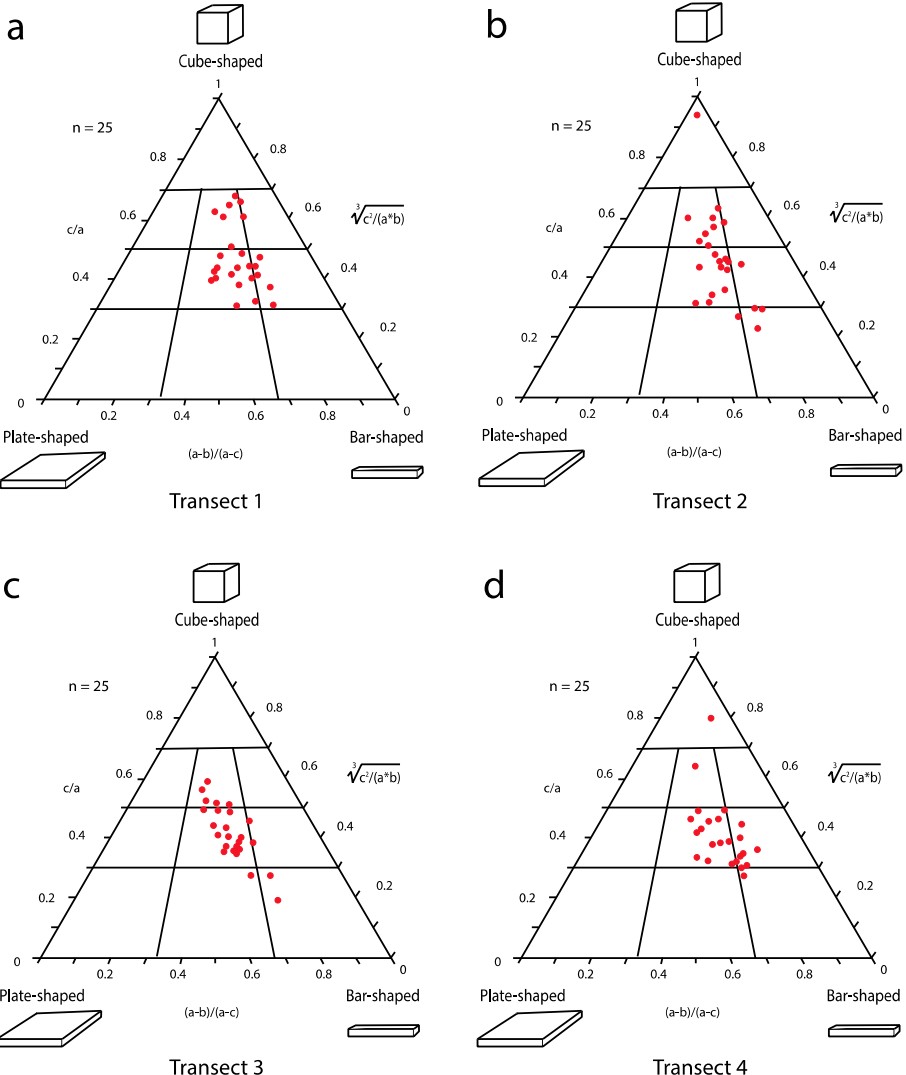

**Figure 7.** Set of triangular Sneed-Folk diagrams used to appraise variations in boulder shape; (**a**) Trend for boulders from Transect 1; (**b**) Trend for boulders from Transect 2; (**c**) Trend for boulders from Transect 3; (**d**) Trend for boulders from Transect 4. Note: All trends slope to the lower right, indicating shapes conserved by elongated boulders.

### 4.6. Analysis of Boulder Sizes

Variations in boulder size as a function of maximum and minimum length drawn from the data sets (Tables A1–A4) may be plotted separately for each transect using bar graphs. Groupings separated by intervals of 25 cm are plotted in histograms stacked to show the trend in diminishing boulder size as a function of distance from the headland source (Figure 8). The largest boulders in the Almeja CBB occur in Transect 1, the distal end of which is 12 m from the closest rhyolite sea cliff (Figure 5). The range in maximum boulder length in Transect 1 is from 58 to 268 cm (Table A1) but the highest frequency falls within the interval of 101 to 125 cm (Figure 8a). The distal end of Transect 2 meets

sea level at a distance of 52 m west along the curve from the bedrock source (Figure 5). The range in maximum boulder length from Transect 2 is from 62 to 172 cm (Table A2) but the largest blocks fall into an interval a full meter less in size than the largest class in Transect 1 (Figure 8b).

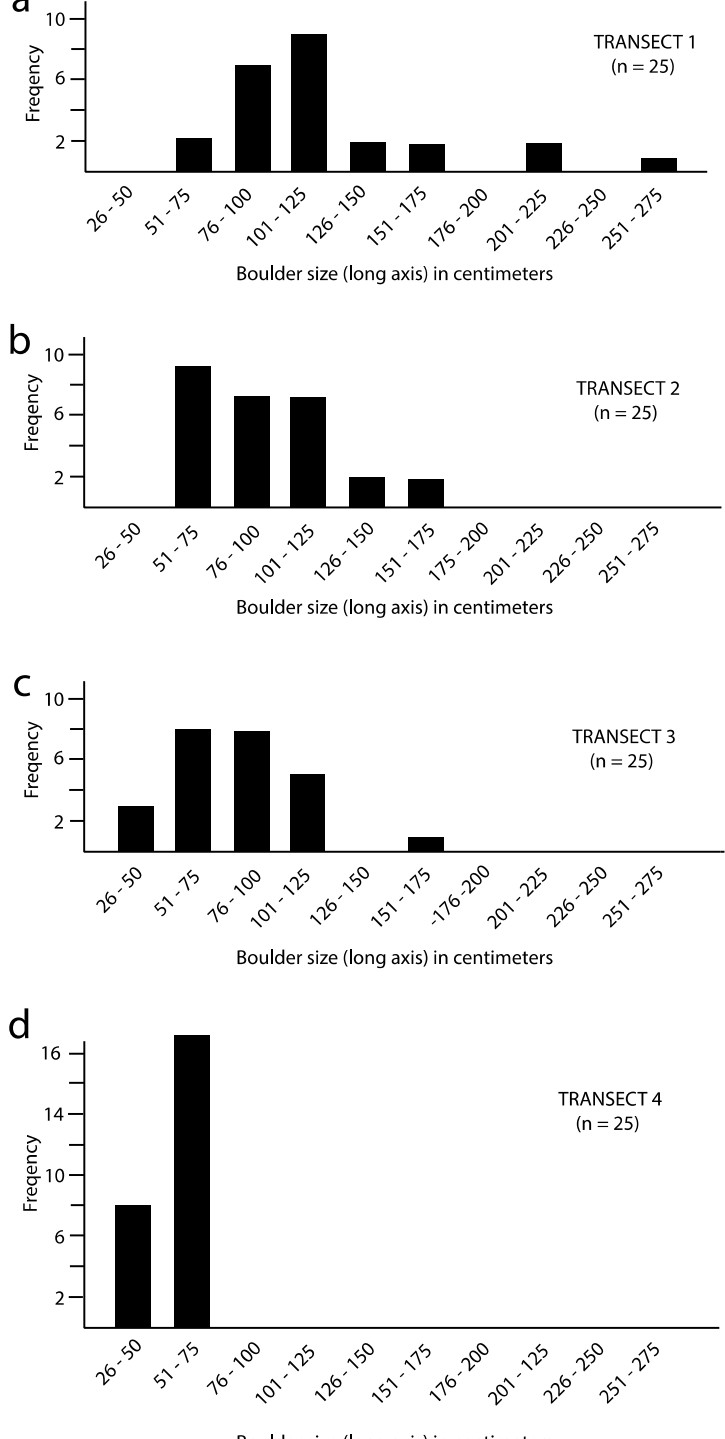

**Figure 8.** Set of bar graphs used to appraise variations in maximum boulder length; (**a**) Size-range and frequency for boulders from Transect 1; (**b**) Same graphic coverage for Transect 2; (**c**) Same graphic coverage for Transect 3; (**d**) Same graphic coverage for Transect 4.

The distal end of Transect 3 joins sea level on the west side of the Almeja CBB, located approximately 160 m down shore from the bedrock source through a curve (Figure 5). The range in maximum boulder length recorded from Transect 3 is from 38 to 164 cm (Table A3) but the largest two populations fall equally into neighboring classes from 51 to 100 cm (Figure 8c). For the most part, boulder populations in transects 2 and 3 overlap in range but the latter includes a smaller population rejected from among the largest 25 samples in all other transects. In contrast, the distal end of Transect 4 intersects sea level on the south margin of the Almeja CBB. At that location, the shore is approximately 230 m along the curve from the same source rocks supplying boulders to the other transect populations (Figure 5). The range in maximum boulder length from Transect 4 is from 33 to 75 cm (Table A4) but by far the largest population occurs within the interval from 51 to 75 cm (Figure 8d).

A similar trend is shown by bar graphs representing minimum boulder length recorded in Tables A1–A4. The range in minimum boulder length from transects 1 and 2 (Figure 9a,b) is significantly greater than found in transects 3 and 4 (Figure 9c,d). By far, the largest populations in transects 1–3 occur in the interval of 26 to 50 cm, although many more clasts in the interval with a maximum size of 25 cm were recorded in Transect 4 (Figure 9d). Clasts with this minimum size are abundant throughout the entire Almeja CBB but were not among the largest 25 samples recorded for minimum length in Transect 1.

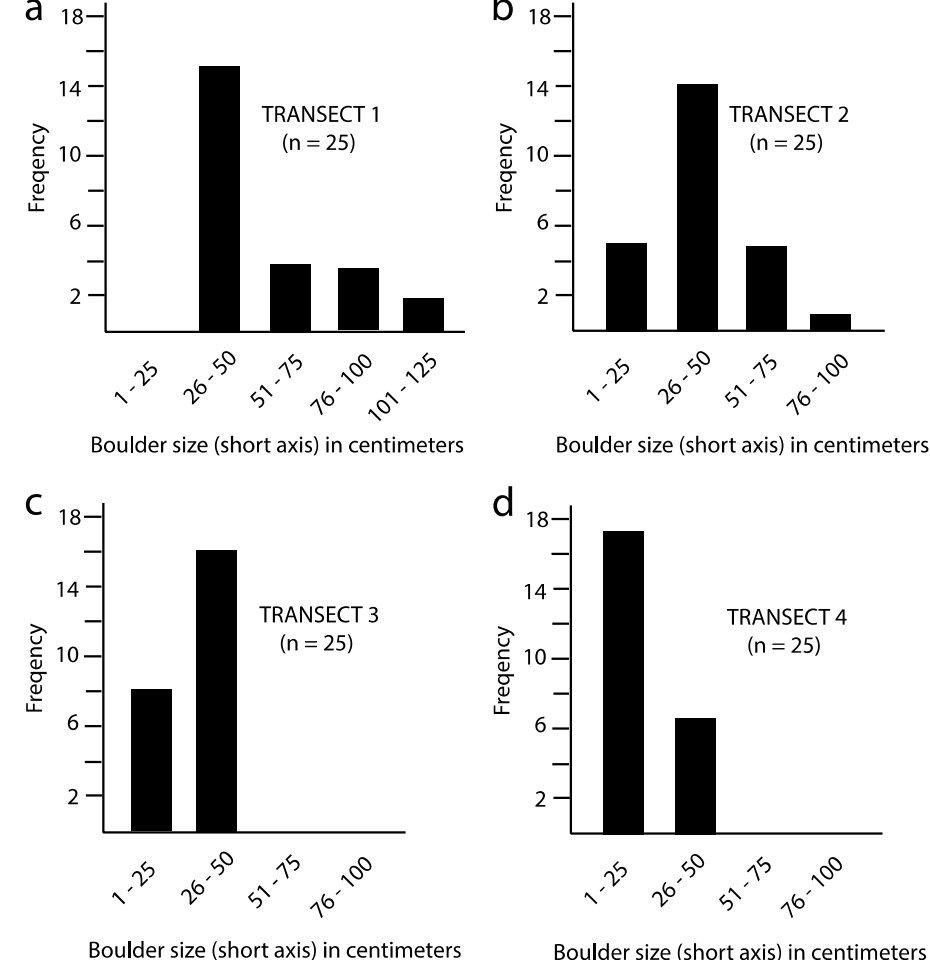

**Figure 9.** Set of bar graphs used to appraise variations in minimum boulder length; (**a**) Size-range and frequency for boulders from Transect 1; Transect 2 (**b**); Transect 3 (**c**); Transect 4 (**d**).

*4.7. Estimation of Wave Heights*

A summary of key data is provided (Table 1), pertaining to average boulder size and maximum boulder size from transects 1 to 4 as correlated with weight calculated on the basis of specific gravity for banded rhyolite. These data are employed to estimate the wave heights required to transport boulders from the bedrock source in sea cliffs to their place in the Almeja CBB. The estimated wave height needed to move the largest boulder encountered in Transect 1 amounts to 13.7 m but that for the largest boulder from Transect 4 is much less at 4.7 m. Estimates using the same equations based on the hydraulic model presented in Section 3.2 are applied to the average boulder weights from Table 1, showing a steady decrease in estimated wave heights from Transect 1 in the most exposed location to Transect 4 in the most sheltered location well within Ensenada Almeja.

Although many of the largest boulders from each transect occur closer to the outer margin and often stand at mean sea level, there are many others that sit well within the CBB. Especially in the western and southern parts of the Almeja CBB, two distinct topographic levels are present. Quantitatively, however, there is no difference in boulder shapes between the inner and outer parts of the deposit. Significantly, the estimated average wave height from the most exposed part of the Almeja CBB on Transect 1 is in agreement with direct observations of wave surge on the rocky shores on the nearby Ensenada San Basilio.

**Table 1.** Summary data from Appendix A (Tables A1–A4) showing maximum boulder size and estimated weight compared to the average values for all boulders (N = 25) from each of transects 1–4 together with calculated values for wave heights estimated as necessary for CBB mobility.

| Tran-Sect | Number of Samples | Average Boulder Size (cm$^3$) | Average Boulder Weight (kg) | Estimated Average Wave ht. (m) | Max. Boulder Size (cm$^3$) | Max. Boulder Weight (kg) | Estimated Wave Height (m) |
|---|---|---|---|---|---|---|---|
| 1 | 25 | 549,340 | 1,201 | 7.9 | 2,264,933 | 4892 | 13.7 |
| 2 | 25 | 182,974 | 395 | 6.0 | 608,546 | 1314 | 8.5 |
| 3 | 25 | 111,118 | 240 | 5.3 | 495,050 | 1069 | 10.3 |
| 4 | 25 | 34,032 | 69 | 3.5 | 111,004 | 240 | 4.7 |

## 5. Discussion

*5.1. Tidal Influence on Coastal Sediment Production*

The most extreme tidal range in the Gulf of California is located in the far north around the delta of the Colorado River, where a macrotidal regime with maximum amplitudes of 12 m plays out over a very low regional slope that results in tidal flats stretching seaward by more than 2 km at low tide [18]. High wave height achieved under wave surge during a major storm that coincides in timing with a high tide can be expected to influence the placement of marine sediments at a higher elevation in any given storm deposit. In the upper gulf region, those sediments are dominated by a peculiar species of marine bivalve (*Mulinia coloradoensis*), the disarticulated shells of which form major intertidal banks called *cheniers* [18]. Tides of this magnitude are unique to the upper part of the gulf. Normally, the daily shift in tidal action during calm weather should have little effect on a particular CBB in the central part of the Gulf of California, where the Guaymas Basin is quite deep. In the case of Ensenada Almeja, the daily tidal cycle does influence the sand beach at the south end of the bay. The modern beach and inland dunes that encroach on the fault valley connected to Ensenada San Basilio [9] are dominated by carbonate sand derived from the abundant bivalve mollusks giving the place its name: Clam Bay. Infaunal bivalves that burrow into the bay's sandy bottom are sheltered during low tide but the disarticulated shells of expired bivalves are liable to be uncovered by currents that accompany changes in the tidal cycle. North-facing sandy beaches in other parts of the central gulf region are commonly enriched by carbonate sand resulting from the breakdown of the abundant bivalve mollusks most commonly belonging to the species *Megapitaria squalida* [8,17].

### 5.2. Seasonal Wind Patterns and Long-Shore Currents

Stiff winds affecting the Gulf of California on an annual basis from November to May [5,7] are capable of generating large-scale wave trains that build in size and travel south over a wide fetch of open water. When sea swells cross into shallow water on approach to a north-facing shoreline, larger waves and surf may be generated that contribute to the abrasion of rocky shores. This action adds finer sediments to local beaches, often enriched by the abrasion of mollusk shells (see above). Such wave activity commonly occurs under clear skies in full sunshine that cannot otherwise be construed as stormy weather. Based on personal boating experience (MEJ), south-directed sea swells with an amplitude of 2 m and wavelength of approximately 10 m are not unusual along the coast near the study site during a wind event lasting several days. In the case of Ensenada Almeja, the waves originating from this source will impact the adjoining, north-facing rocky headland. Some amount of erosion can be expected against the exposed bedrock. However, this particular energy source does not account for the strikingly asymmetrical configuration of the Ensenada Almeja CBB. As apparent from aerial surveillance (Figure 3), the amount of boulder debris eroded from the east side of the headland is insignificant compared to the partial-ring CBB accumulated on the west side. Long-shore currents generated by the seasonal north winds in the region are insufficient to account for the geomorphology of the Almeja CBB.

### 5.3. Extra-regional Tsunami Activity

Deposits correlated with extreme wave action in other parts of the Mexico's Pacific coastline have raised the question as to whether a source from tsunamis can be distinguished from major storms [19]. In particular, the adjoining Jalisco and Michoacán states of Mexico far south of Baja California Sur are bordered by an active subduction zone resulting from compression between the Rivera lithospheric plate and the continental mainland. Among the historical events recorded for this region, the 22 June 1932 earthquake (magnitude 7.7) was one of the region's most destructive affecting an area 1 km inland along a 20-km stretch of coast with a run-up of 15 m [20]. In contrast, the Gulf of California has no historical record of tsunami events, although shallow earthquakes are relatively common due to transtensional tectonics associated with activity along multiple strike-slip faults that dissect narrow sea-floor spreading zones [5]. Traces of former subduction zones related to the San Benito and Tosco-Abreojos faults extend offshore along much of the outer Pacific coast of Baja California but these ceased to be active approximately 12 million years ago [5]. Rocky shores along the inner gulf coast of the Baja California peninsula close to the study site entail steep cliffs that rise abruptly to elevations as high as 100 m (Figure 1c). Nothing has been described as remotely similar to the Pleistocene tsunami deposits documented with a run-up of 270 m against the steep volcanic shores of Santiago in the Cape Verde Islands [21]. The hypothesis of a tsunami origin for the Ensenada Almeja CBB is easily eliminated on account of the barren zone lacking boulders inside the partial-ring construction (Figure 5), as well as the occurrence of the CBB restricted to one side of Ensenada Almeja. Any potential tsunami source would have filled the interior of the half-ring with boulders. Moreover, comparable deposits can be expected to have formed along both sides of the bay.

### 5.4. Hurricane Frequency

As many as 25 to 30 tropical depressions originate each year near Acapulco off western Mexico between the months of May and November before intensifying in strength and shifting northwest into the eastern Pacific Ocean [22]. Especially during El Niño events every 6 to 8 years, a few storms turn northward into the Gulf of California. In recent years, several hurricanes have struck the southern tip of the Baja California peninsula and followed tracks crossing Isla Cerralvo in the southern gulf region, where heavy rainfall flushes arroyo sediments to form tide-water deltas at some 39 localities around the island's circumference [23]. Long-shore currents stimulated by the winter winds also play a role in truncating those deltas and sending the sediment load south along both sides of the island. Hurricanes that manage

to enter the gulf usually lose energy rapidly before continuing as downgraded storms, although Hurricane Odile is a recent exception, reaching Loreto at hurricane strength in 2014. The driest part of the Baja California peninsula is located far to the north in the upper Gulf of California, where normal rainfall amounts to only 5 cm per year. A remnant of Hurricane Odile was the last big storm to bring excess water to the area. In particular, the large Costilla Delta that empties sediments from Heme Canyon south of Puertecitos (Figure 1a) lends evidence to the effect of episodic rain storms that flush the region, although a nearby salt lagoon also attests to long periods of aridity [24]. The bar that closes off the salt lagoon was constructed under the influence of long-shore currents based on the occurrence of pumice cobbles derived from strata within Heme Canyon and transferred seaward via the Costilla Delta.

Although clearly episodic in frequency and less common in the northern Gulf of California, hurricanes are the major factor capable of expending sufficient energy to shape the landscapes of peninsular Baja California through the agencies of stream erosion and shore modification. The Almeja CBB, in particular, stands out as a prime example of a large but distinctly asymmetrical deposit that only could have been formed under the influence of incremental additions due to Holocene hurricanes with a counter-clockwise rotation sending wave surge westward across the headland.

### 5.5. Human Occupation of the San Basilio Area

Archeological evidence of kitchen middens including worked flakes of obsidian occurs on the northwest shore of Ensenada San Basilio in one of the most sheltered corners of the bay. Cave paintings also are known from a locality on the south side of the bay. These remains indicate that the area has a history of occupation predating the arrival of Europeans. No trace of habitation is known from nearby Ensenada Almeja, possibly because of exposure to the seasonal north winds. Nonetheless, native peoples would have been subjected to storm conditions from time to time.

### 5.6. Regional Patterns for Coastal Boulder Beds

Study of rocky-shore attrition around the Gulf of California due to impact by hurricanes through the last 10,000 years has barely commenced with the only previous example based on the limestone CBB on the east coast of Isla del Carmen [2]. The largest up-turned blocks of layered limestone from the Carmen CBB are estimated to weigh between 5.8 and 28 metric tons. The largest megaclast in the Almeja CBB (Table A1, Transect 1) is close to 5 metric tons in weight. Approximately 30% of the megaclasts measured from Transect 1 exceed one metric ton in weight. By comparison, only two of the megaclasts from the next transect (Table A2, Transect 2) exceed one metric ton in weight and only one from the third transect (Table A3) exceeds that amount. None of the boulders in the last transect (Table A4) come close to a metric ton.

To what extent might other examples of Holocene or older Pleistocene CBBs exist throughout the Gulf of California and what source rocks are most typically represented? The Loreto area offers additional possibilities for expanded studies. Located 23 km south of Loreto (Figure 1b), Puerto Escondido is a natural harbor with a single entrance from the southeast leading to a large inner lagoon sheltered by islets linked by boulder barriers eroded from an adjacent headland (El Chino) at one end and the largest island (La Enfermeria) at the other. Overall, the andesite clasts on the barriers are poorly sorted with a wide range of sizes similar to the Almeja CBB. Future research may determine to what extent the barriers were formed by long-shore currents or a combination of factors including storm activity.

A short distance north of Loreto, the south shore of Isla Coronados (Figure 1b) is clad by andesite boulders forming an extensive berm. Given that the south shore is on the leeward side of the island sheltered from the north winds and related sea swell, the berm is more likely to have been activated by hurricane activity. In addition, a Pleistocene lagoon inland from the unconsolidated boulder berm is filled with limestone that dips northward away from a bedrock ridge with the internal carbonate layers interpreted as over-wash deposits derived from rhodolith debris [25]. The most likely mechanism for north-directed over-wash events on Isla Coronados would have resulted from major storms or hurricanes arriving from the south.

A fitting analog for the Almeja CBB is the 400-m long paleoshore near Punta San Antonio (Figure 1b), formed by mixed granodiorite and andesite boulders to which a diverse Pleistocene biota is attached in growth position [26]. In particular, granodiorite boulders derive from the adjacent headland at Punta San Antonio that occupies a flanking position comparable to the rhyolite headland at Ensenada Almeja. The paleogeography of the Punta San Antonio site also features a former embayment comparable in size to Ensenada Almeja. As there is no bedrock exposure of granodiorite north of the former bay, wind-driven currents from that direction could not have been responsible for development of the Pleistocene CBB. The only alternative is an energy source associated with the passage of Pleistocene hurricanes with a counter-clockwise rotation suited to erosion of the Punta San Antonio headland to the east.

Another area with rich potential for future studies on CBBs is located in the upper Gulf of California off Bahía Los Angeles (Figure 1a). The southeast end of Isla Angel de la Guarda is known for its closed lagoons with elevated salinities that favor living microbial colonies [27], commonly recognized by paleontologists and geologists as stromatolites. Based on personal exploration (MEJ and JL-V), the smaller lagoon on Isla Angel de la Guarda (Figure 10, number 1) is closed off by a CBB formed by large andesite boulders. The principal source for this material is the adjacent rocky shore to the north, which features eroded sea stacks. As long-shore currents from the north are blocked by nearby Isla Estanque, the alternative energy source for erosion of the andesite cliffs close to the small lagoon is likely to have been the result of episodic storms or hurricanes. Based again on personal experience, andesite clasts on the deposit closing off the larger lagoon on Angel de la Guarda (Figure 10, number 2) are mostly the size of cobbles. Long-shore drift may have been more constructive in the development of the enclosing berm. Isla Estanque has yet to be explored with any focus on lagoon development (Figure 10, numbers 3 and 4) but the clock-wise rotation of storm systems offers a promising hypothesis for the development of boulder spurs yet to completely isolate related lagoons.

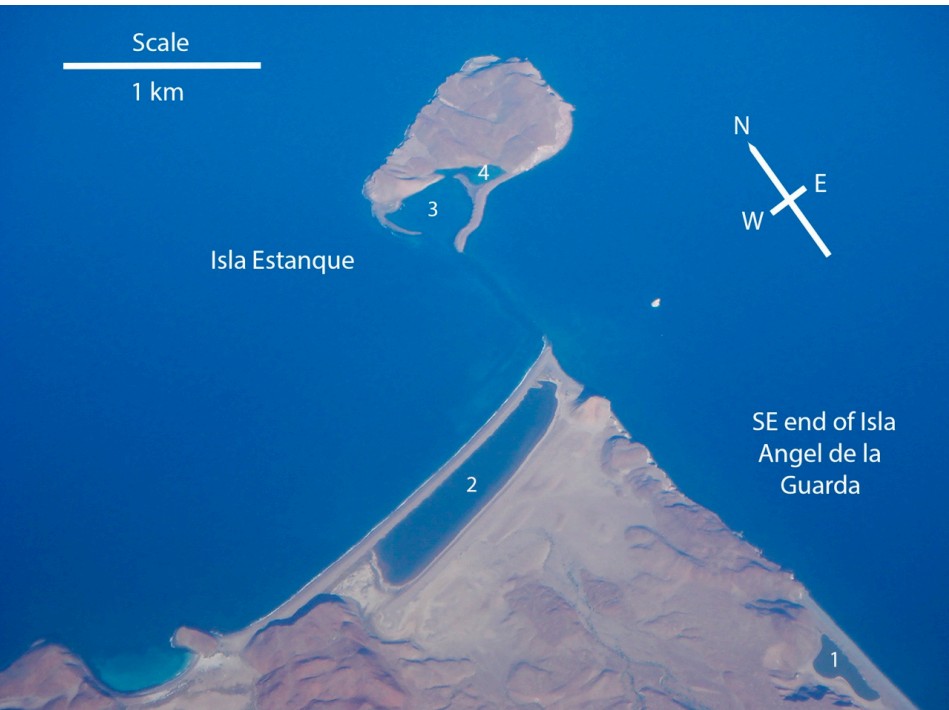

**Figure 10.** Aerial photo from an altitude of 15,000 m showing boulder deposits from the southeast coast of Isla Angel de la Guarda in the upper Gulf of California that define closed lagoons (1 and 2) and distinct spurs formed by bolder deposits on Isla Estanque in the process of closing other lagoons (3 and 4).

*5.7. Comparison to Selected CBBs Elsewhere in the World*

Ruban et al. (2019) compiled a representative collection of 58 published studies concerning Earth-bound processes capable of producing megaclasts [11]. From this sample, more than half (53%) are limited to boulder deposits that formed during Quaternary time, 20 of which represent coastal depositional settings. The data base distinguishes between Quaternary CBBs attributed to storms as opposed to tsunamis in nearly equal parts. A clear-cut example of a huge tsunami event derives from the study by Ramalho et al. (2015) with regard to basalt megaclasts as much as 8 m in diameter with a maximum estimated weight of as much as 1000 metric tons, left high on the flanks of Santiago in the Cape Verde Islands [21]. An equally clear-cut study by Cox et al. (2018) relates to blocks with an estimated weight as much as 620 metric tons pealed back from Carboniferous limestone layers exposed at the top of high sea cliffs in western Ireland [28] that are unequivocally linked to major sea storms. Not included in the data base of Ruban et al. (2019) are other studies on massive carbonate megaclasts from the Bahamas and Bermuda interpreted ambiguously as either tsunami or storm-related [29] or unequivocally as storm related [30]. Current literature on Quaternary CBBs appears to be skewed towards studies on carbonate megaclasts, such as the work by Biolchi et al. (2019) from the northern Adriatic Sea [31]. Bedded limestone formations exposed in sea cliffs are especially vulnerable to erosion by storm-induced waves, as exemplified by our previous study on a Holocene CBB from Isla del Carmen in the Gulf of California [2]. Volcanic flows composed of layered basalt and andesite also form extensive sea cliffs around the Gulf of California [3] and many other parts of the world. The geomorphology of CBBs stripped from igneous basement rocks is underrepresented in the literature and offers a research target worthy of future investigations especially in the context of likely hurricane deposits.

## 6. Conclusions

- Hurricanes strike Mexico's Baja California peninsula and enter the Gulf of California with increased frequency especially during El Niño years commonly repeated every 6 to 8 years. The last hurricane known to reach Ensenada Almeja in the San Basilio area north of Loreto was Hurricane Odile in 2014.

- By process of elimination taking into account more frequent but less energetic sources of input such as tidal forces, seasonal wind patterns involved with long-shore currents, as well as tsunamis, data are found to favor an incremental accumulation of the Ensenada Almeja CBB due to a repetition of hurricane events through Holocene time.

- Maximum wave height stimulated by a major storm necessary to erode the largest blocks of banded rhyolite with a calculated specific gravity of 2.16 is estimated to have been on the order of 13.7 m.

- Evidence based on size distribution in boulders from four different transects crossing perpendicular through the Almeja CBB shows a decrease in maximum size along a curved shoreline ending 230 m distal from the bedrock source on the outer tip of the adjacent headland. Loss of energy is due to wave refraction entering Ensenada Almeja after impact against the headland with wave surge arriving from the east driven by a counter-clockwise rotation of a hurricane system.

- The restriction of embayments by CBBs and the related closure of lagoons by boulder spurs in the form of unconsolidated bars, derived from bedrock sources of basalt and andesite, is a widespread pattern in the Gulf of California. Future efforts that distinguish between different energy sources related to coastal erosion of exposed bedrock must take into consideration the importance of hurricanes and down-graded tropical storms that impact the region on an episodic basis. Like rhyolite (this study), basalt and andesite are susceptible to intense hydrologic pressure exerted against parting seams and vertical joints exposed to wave action during major storms.

**Supplementary Materials:** The following video is available online at http://www.mdpi.com/2077-1312/7/6/193/s1, Video Hurricane Odile at San Basilio.mov.

**Author Contributions:** Initial field reconnaissance was conducted by M.E.J. and J.L.-V. in March 2017 with a follow-up visit by J.L.-V. in March 2018 paying close attention to patterns of natural weathering in the rhyolitic basement rocks exposed on the headland at Ensenada Almeja. Fieldwork resulting in collection of shape and size data at the Almeja CBB was carried out in May 2019 by M.E.J. and E.M.J. M.E.J. prepared the first draft of this contribution, drafted all figures and supplied all ground photos. R.G.-F. was responsible for working out the mathematics related to storm hydrodynamics.

**Funding:** This research received no external funding.

**Acknowledgments:** Foremost, we are indebted to Norm Christy, part-time resident of Loreto, for his invaluable assistance with logistics during our 2019 visit and for launching his DJI Phantom-2 drone to provide aerial photos of the Ensenada Almeja CBB. Eric Stevens provided critical insight with his video of storm action at San Basilio during Hurricane Odile. Special thanks are due to Tom Woodard in Loreto for arranging our stay at the Spanish Contessa's house at the conclusion of the project. M.E.J. is grateful to Jay Racela (Environmental Lab, Williams College) for help with the experimental calculation of density for the banded rhyolite sample from Ensenada Almeja. Reviews of an earlier manuscript for which the authors are most grateful were provided by two anonymous readers, as well as by Dmitry A. Ruban (Geology and Geography Faculty, Southern Federal University, Russia).

**Conflicts of Interest:** The authors declare no conflict of interest.

## Appendix A

**Table A1.** Quantification of boulder size, volume and estimated weight from CBB samples through Transect 1 at Ensenada Almeja. The laboratory result for density of banded rhyolite at 2.16 gm/cm$^3$ is applied uniformly to all samples in this table.

| Sample | Distance to Next (cm) | Long Axis (cm) | Intermediate Axis (cm) | Short Axis (cm) | Volume (cm$^3$) | Adjust. to 65% | Weight (kg) | Estimated Wave ht. (m) |
|---|---|---|---|---|---|---|---|---|
| 1 | 0 | 88 | 44 | 38 | 147,136 | 95,638 | 207 | 5.5 |
| 2 | +20 | 58 | 40 | 36 | 83,520 | 54,288 | 219 | 3.6 |
| 3 | +60 | 108 | 60 | 44 | 285,120 | 185,328 | 296 | 6.8 |
| 4 | +90 | 70 | 64 | 47 | 210,560 | 136,864 | 647 | 4.4 |
| 5 | +104 | 110 | 81 | 523 | 463,320 | 301,158 | 651 | 6.9 |
| 6 | +320 | 116 | 74 | 50 | 429,200 | 278,980 | 603 | 7.3 |
| 7 | +108 | 112 | 82 | 45 | 413,280 | 268,632 | 580 | 7.0 |
| 8 | +100 | 92 | 52 | 43 | 205,712 | 133,713 | 289 | 5.8 |
| 9 | +100 | 155 | 115 | 93 | 1,657,725 | 1,077,521 | 2327 | 9.7 |
| 10 | +140 | 92 | 88 | 60 | 485,760 | 315,744 | 682 | 5.8 |
| 11 | +208 | 208 | 118 | 78 | 2,470,624 | 1,605,906 | 3469 | 13.0 |
| 12 | +330 | 120 | 56 | 50 | 336,000 | 218,400 | 472 | 7.5 |
| 13 | +220 | 268 | 111 | 104 | 3,093,792 | 2,010,965 | 4344 | 16.8 |
| 14 | +150 | 97 | 74 | 30 | 215,340 | 139,971 | 302 | 6.1 |
| 15 | +100 | 153 | 126 | 98 | 1,889,244 | 1,228,009 | 2653 | 9.6 |
| 16 | +150 | 108 | 84 | 44 | 399,168 | 259,459 | 560 | 6.8 |
| 17 | +80 | 105 | 60 | 33 | 207,900 | 135,135 | 292 | 6.6 |
| 18 | +0 | 92 | 80 | 34 | 250,240 | 162,656 | 351 | 5.8 |
| 19 | +120 | 125 | 118 | 75 | 1,106,250 | 719,063 | 1553 | 7.8 |
| 20 | +0 | 87 | 66 | 38 | 218,196 | 141,827 | 306 | 5.5 |
| 21 | +100 | 130 | 121 | 60 | 943,800 | 613,470 | 1325 | 8.2 |
| 22 | +200 | 214 | 94 | 85 | 1,709,860 | 1,111,409 | 2401 | 13.4 |
| 23 | +0 | 78 | 62 | 334 | 164,424 | 106,876 | 231 | 4.9 |
| 24 | +100 | 128 | 53 | 38 | 257,792 | 167,565 | 362 | 8.0 |
| 25 | +100 | 218 | 148 | 108 | 3,484,512 | 2,264,933 | 4892 | 13.7 |
| Average | +158 | 125 | 83 | 57 | 845,139 | 549,340 | 1201 | 7.9 |

**Table A2.** Quantification of boulder size, volume and estimated weight from CBB samples through Transect 2 at Ensenada Almeja. The laboratory result for density of banded rhyolite at 2.16 gm/cm$^3$ is applied uniformly to all samples in this table.

| Sample | Distance to Next (cm) | Long Axis (cm) | Intermediate Axis (cm) | Short Axis (cm) | Volume (cm$^3$) | Adjust. to 65% | Weight (kg) | Estimated Wave ht. (m) |
|---|---|---|---|---|---|---|---|---|
| 1 | 0 | 83 | 63 | 24 | 125,496 | 81,572 | 176 | 5.2 |
| 2 | +130 | 62 | 31 | 60 | 30,752 | 19,989 | 43 | 3.9 |
| 3 | +220 | 74 | 45 | 33 | 49,728 | 32,323 | 70 | 4.6 |
| 4 | +200 | 66 | 50 | 34 | 98,010 | 63,707 | 138 | 4.1 |
| 5 | +600 | 172 | 67 | 46 | 122,400 | 79,560 | 172 | 10.8 |
| 6 | +140 | 86 | 69 | 21 | 284,832 | 185,141 | 400 | 5.4 |
| 7 | +260 | 68 | 27 | 26 | 38,556 | 25,061 | 54 | 4.3 |
| 8 | +150 | 75 | 35 | 25 | 65,626 | 42,656 | 92 | 4.7 |
| 9 | +700 | 64 | 48 | 29 | 89,089 | 57,907 | 125 | 4.0 |
| 10 | +150 | 128 | 42 | 39 | 209,664 | 136,282 | 294 | 8.0 |
| 11 | +130 | 61 | 56 | 38 | 129,808 | 84,375 | 182 | 3.8 |
| 12 | +900 | 98 | 81 | 28 | 222,264 | 144,472 | 312 | 6.2 |
| 13 | +200 | 92 | 58 | 55 | 293,480 | 190,762 | 412 | 5.8 |
| 14 | +0 | 74 | 51 | 33 | 124,542 | 80,952 | 175 | 4.6 |
| 15 | +800 | 108 | 64 | 38 | 262,656 | 170,726 | 369 | 6.8 |
| 16 | +310 | 115 | 83 | 48 | 458,160 | 297,804 | 643 | 7.2 |
| 17 | +10 | 125 | 85 | 53 | 563,125 | 336,031 | 726 | 7.8 |
| 18 | +200 | 108 | 85 | 48 | 440,640 | 286,416 | 619 | 6.8 |
| 19 | +100 | 106 | 55 | 45 | 262,350 | 170,528 | 368 | 6.7 |
| 20 | +220 | 113 | 71 | 58 | 465,334 | 302,467 | 653 | 7.1 |
| 21 | 0 | 135 | 95 | 73 | 936,225 | 608,546 | 1314 | 8.5 |
| 22 | +330 | 83 | 78 | 48 | 310,752 | 201,993 | 436 | 5.2 |
| 23 | +250 | 88 | 73 | 88 | 234,048 | 217,131 | 469 | 5.5 |
| 24 | +40 | 94 | 91 | 19 | 752,752 | 489,293 | 1057 | 5.9 |
| 25 | +60 | 101 | 93 | 44 | 413,292 | 268,640 | 583 | 6.3 |
| Average | 100 | 91 | 63 | 41 | 283,343 | 182,974 | 395 | 6.0 |

**Table A3.** Quantification of boulder size, volume and estimated weight from CBB samples through Transect 3 at Ensenada Almeja. The laboratory result for density of banded rhyolite at 2.16 gm/cm$^3$ is applied uniformly to all samples in this table.

| Sample | Distance to next (cm) | Long axis (cm) | Intermediate axis (cm) | Short axis (cm) | Volume (cm$^3$) | Adjust. to 65% | Weight (kg) | Estimated Wave ht. (m) |
|---|---|---|---|---|---|---|---|---|
| 1 | 0 | 79 | 38 | 21 | 63,042 | 40,977 | 89 | 5.0 |
| 2 | +400 | 76 | 53 | 38 | 153,064 | 99,492 | 215 | 4.8 |
| 3 | +100 | 64 | 34 | 22 | 47,872 | 31,117 | 67 | 4.0 |
| 4 | +260 | 54 | 32 | 23 | 39,744 | 25,834 | 56 | 3.4 |
| 5 | +200 | 72 | 40 | 26 | 74,880 | 48,672 | 105 | 4.5 |
| 6 | 0 | 45 | 37 | 20 | 33,300 | 21,645 | 47 | 2.8 |
| 7 | +120 | 38 | 25 | 15 | 14,250 | 9263 | 20 | 2.4 |
| 8 | +800 | 65 | 36 | 23 | 53,820 | 34,983 | 76 | 4.1 |
| 9 | +270 | 46 | 25 | 18 | 20,700 | 13,455 | 29 | 2.9 |
| 10 | +230 | 109 | 53 | 20 | 115,540 | 75,101 | 162 | 6.8 |
| 11 | +800 | 111 | 56 | 48 | 298,363 | 193,939 | 419 | 7.0 |
| 12 | +130 | 100 | 50 | 48 | 240,000 | 156,000 | 337 | 6.3 |
| 13 | +220 | 66 | 37 | 34 | 83,028 | 53,968 | 117 | 4.1 |
| 14 | +230 | 75 | 36 | 27 | 72,900 | 47,385 | 102 | 4.7 |
| 15 | +40 | 69 | 40 | 38 | 104,880 | 68,178 | 147 | 4.3 |
| 16 | +20 | 89 | 55 | 45 | 220,275 | 143,179 | 309 | 5.6 |
| 17 | +170 | 99 | 43 | 34 | 144,738 | 94,080 | 203 | 6.2 |
| 18 | +240 | 76 | 48 | 44 | 160,512 | 104,333 | 225 | 4.8 |

**Table A3.** *Cont.*

| Sample | Distance to next (cm) | Long axis (cm) | Intermediate axis (cm) | Short axis (cm) | Volume (cm³) | Adjust. to 65% | Weight (kg) | Estimated Wave ht. (m) |
|--------|-----------------------|----------------|------------------------|-----------------|--------------|----------------|-------------|------------------------|
| 19 | +120 | 75 | 56 | 28 | 117,600 | 76,440 | 165 | 4.7 |
| 20 | +60 | 83 | 50 | 40 | 166,000 | 107,900 | 233 | 5.2 |
| 21 | +220 | 120 | 58 | 48 | 334,080 | 217,152 | 469 | 7.5 |
| 22 | 0 | 164 | 108 | 43 | 761,616 | 495,050 | 1069 | 10.3 |
| 23 | +40 | 108 | 58 | 38 | 238,032 | 3,154,721 | 334 | 6.8 |
| 24 | +180 | 92 | 63 | 44 | 255,024 | 165,766 | 358 | 5.8 |
| 25 | +140 | 123 | 78 | 48 | 460,512 | 299,333 | 647 | 7.7 |
| Average | 140 | 84 | 48 | 33 | 170,951 | 111,118 | 240 | 5.3 |

**Table A4.** Quantification of boulder size, volume and estimated weight from CBB samples through Transect 4 at Ensenada Almeja. The laboratory result for density of banded rhyolite at 2.16 gm/cm³ is applied uniformly to all samples in this table.

| Sample | Distance to Next (cm) | Long Axis (cm) | Intermediate Axis (cm) | Short Axis (cm) | Volume (cm³) | Adjust. to 65% | Weight (kg) | Estimated Wave ht. (m) |
|--------|-----------------------|----------------|------------------------|-----------------|--------------|----------------|-------------|------------------------|
| 1 | +700 | 50 | 30 | 19 | 28,500 | 18,525 | 40 | 3.1 |
| 2 | 0 | 48 | 18 | 16 | 13,824 | 89,896 | 19 | 3.0 |
| 3 | +300 | 33 | 27 | 13 | 11,583 | 7529 | 16 | 2.1 |
| 4 | +170 | 53 | 36 | 20 | 38,160 | 24,804 | 54 | 3.3 |
| 5 | +160 | 40 | 25 | 12 | 12,000 | 7800 | 17 | 2.5 |
| 6 | +190 | 44 | 27 | 14 | 16,632 | 10,810 | 23 | 2.8 |
| 7 | +800 | 42 | 41 | 15 | 25,830 | 16,790 | 36 | 2.6 |
| 8 | +600 | 38 | 38 | 30 | 43,320 | 28,158 | 61 | 2.4 |
| 9 | +200 | 59 | 31 | 22 | 40,238 | 26,155 | 56 | 3.7 |
| 10 | +110 | 52 | 30 | 25 | 39,000 | 23,350 | 50 | 3.3 |
| 11 | 0 | 62 | 44 | 19 | 51,832 | 33,691 | 73 | 3.9 |
| 12 | +700 | 41 | 33 | 20 | 27,060 | 17,589 | 38 | 2.6 |
| 13 | 0 | 53 | 38 | 18 | 36,252 | 23,564 | 51 | 3.3 |
| 14 | 0 | 61 | 43 | 28 | 73,444 | 47,739 | 103 | 3.8 |
| 15 | +900 | 63 | 32 | 29 | 58,464 | 38,002 | 82 | 4.0 |
| 16 | 0 | 67 | 48 | 42 | 135,072 | 87,797 | 190 | 4.2 |
| 17 | +100 | 75 | 69 | 33 | 170,775 | 111,004 | 240 | 4.7 |
| 18 | +20 | 71 | 30 | 23 | 48,990 | 31,844 | 69 | 4.5 |
| 19 | 0 | 58 | 33 | 18 | 34,452 | 22,394 | 48 | 3.6 |
| 20 | +500 | 66 | 35 | 28 | 64,680 | 42,042 | 91 | 4.1 |
| 21 | +200 | 60 | 42 | 20 | 50,400 | 32,760 | 71 | 3.8 |
| 22 | 0 | 59 | 35 | 16 | 33,040 | 21,476 | 46 | 3.7 |
| 23 | 0 | 68 | 33 | 28 | 62,832 | 40,841 | 88 | 4.3 |
| 24 | +400 | 58 | 43 | 20 | 49,880 | 32,422 | 70 | 3.6 |
| 25 | +20 | 63 | 39 | 28 | 68,796 | 44,717 | 97 | 4.0 |
| Average | 100 | 55 | 36 | 22 | 49,402 | 34,032 | 69 | 3.5 |

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
