# Peer review of "Geomorphology of a Holocene Hurricane Deposit Eroded from Rhyolite Sea Cliffs on Ensenada Almeja (Baja California Sur, Mexico)"

_jmse, doi:10.3390/jmse7060193_

Round 1
Reviewer 1 Report
I revised the manuscript and I have general and specific comments.
General comments. The quality of English is quite good. The first part of the manuscript as well as the description of “geological” results, i.e. boulders’ sampling, description of size, location, etc. is well done. Unfortunately it is not the same (in my opinion) for the second part that should give qualitative data on winds, sea waves, swell waves, storm waves and hurricane waves….the same for tides…no information at all is presented. In “Discussion”, results obtained have to be compared with other similar studies…..I am also sorry to state that the comparison with other localities is a bit confused because such localities are not presented in maps and they have not been previously described… …
Specific comments.
Lines 14-16: “A previous contribution looked at evidence for the formation of 14 a coastal boulder bed (CBB) formed by enormous blocks of limestone pealed back from the edge of 15 a marine terrace on an island off Loreto in Baja California Sur” and lines 25-26 “Additional…” ….I suggest to erase both sentences since the abstract has to show only results of the present study.
Line 19 (and line 61), is “Shell” Bay? Or “Clam” Bay?
Line 55, there is a parenthesis that has to be checked.
Line 76 probably is “that” and not “which”.
Line 77 is “yearly” value I guess…please add and I would mind t9o suggest give values in “mm” not “cm”.
Line 84, I guess this is the place and moment in which you have to give more quantitative information about tides and winds since you use them later in “Discussion”.
Line 86. This “S25°E” rounds a bit strange to me….probably would be better to express that from N to E as you do at line 103. The same for the other value at line 103.
In Figure 1 a I would suggest to add a box indicating the study area position…
Lines 135-142. I would suggest to number equations and Hs I guess is for significant wave height….
Lines 157-161. Probably this issue should go to discussion …..
Lines 245 etc…I guess it is not necessary to repeat “cm” each time…that is: from 58 to 268 cm….
Line 247, I suggest add “of” 52 m…..further: “due west from” is not clear, please explain.
Figure 8, it is not necessary to say “cm” each time. The same for figure 9.
Line 265, the word “treatment” is probably not correct…better say “trend”…
Line 267, “that found”
Line 268, “cm” is repeated two times…
Line 269, “transect”
Figure 9. “(a) Size-range and frequency for boulders from Transect 1; (b) Same graphic coverage for Transect 2; (c) Same graphic coverage for Transect 3; (d) Same graphic coverage for Transect 4.” It could be re-phrased like this:
“Size-range and frequency for boulders from Transect 1 (a), transect 2 (b)……
Line 282, I guess you should mention equations’ numbers.
Line 285, “the” is probably missing before Ensenada …
Line 305 I suggest to add the scientific name.
Line 321, numerical, quantitative data are required…..if not everything is too simple and qualitatively described…anyway it is quite obvious (I guess) that both tides and wind waves have no role in boulders’ transport..
Line 325, “disturbances”? maybe “depressions”?
Line 326, please give months names
Lines 329 and 338, you mention localities that are not in the figures….
Line 348, You can mention data on bolder weigth…..etc..and especially compare them with other studies...
Lines 351-352, not clear….
Line 370, “on is on” is probably a mistake…
Line 390 (and 395), “based on personal experience”…of whom?
Author Response
From the general comments of Reader #1, the first part of our ms. through the results section was rated as “well done.” However, the second part of the paper which involves the discussion section was found to be lacking and in need of major revisions. Specifically, “qualitative data on winds, sea waves, swell waves, storm waves and hurricane waves, and tides” was requested.
As for hurricane waves, I have now submitted to the managing editor (Yuri Wang) a copy of the video we’d like to see adopted as Supplementary Material. The video is cited in the text, but the file was so large that I was unable to send it during the initial submission stage. Getting a firm handle on hurricane waves is not easy, but based on the video made at a locality very close to our study site, we are able to say that the run-up of hurricane waves against the rocky shore amounts to 8 m. This value agrees very well with the median wave size arrived at through our mathematical formula.
Appropriate citations from the literature are cited with regard to wind speed for the Winter North winds on the Gulf of California (which we specifically note are NOT storm related). Same goes for tides. Regarding sea swells during the Winter winds, I have added my personal observations made during boat runs in the general area of the study site. Small boat traffic is impossible going north during a big blow, but south-moving traffic is quite easy in the direction of the moving swells and a 2-m difference between troughs and highs is the extreme. The last item mentioned with reference to specific lines in the text, asks to state by whom the “personal experience” is cited. This information has been added where ever appropriate, showing my initials and/or those of my Mexican field partner and co-author on this paper (Jorge Ledesma-Vázquez). We have 30 year’s experience working together in Baja California and many of these direct observations are cited in our published papers.
Reader #1 also asks that we compare the results of our study with other similar studies. Ours is only the second study of its kind to be done in the Gulf of California. Specific information has now been added to the discussion section comparing the results of our present study (rhyolite boulders) to our previous 2018 study in JMSE at another locality where limestone boulders form the storm deposit. Our present study and the 2018 study are the ONLY studies of their kind anywhere in the Gulf of California.
As part of the discussion, we specifically mention several other localities where this kind of study may be expanded to include other volcanic rocks such as basalt and andesite. In all cases, these localities are clearly marked in the various maps from Figure 1. Care was taken to amend Figure 1 to make sure this is the case.
The discussion section now also includes comparisons with other coastal boulder beds from elsewhere around the world (in response to Reader #2).
In the second part of the review by Reader #1, 26 specific recommendations were made tied to specific lines in the original draft of the paper. As mentioned (above), the last of these was satisfied by added author’s initials to references about “personal experience.”
I am able to respond that all but one of these suggestions tied to specific lines is now accommodated in the revised text.
This also entailed modifications to Figures 1, 8, and 9 following the very specific instructions of the reviewer.
The only exception is from lines 157-161, where we justify results in determining the specific gravity of our rhyolite sample with comparisons to other published work. The comment was: “Probably this issue should go to discussion.” We disagree and wish to maintain the text as is, in this particular case.
Otherwise, every attempt was made to accommodate all other recommendations offered by Reader #1. His/her report was the longest and most detailed review of the three obtained for this project. We are very grateful for the careful attention paid to our work and acknowledgement has been added to the text.
Reviewer 2 Report
This is a very interesting paper that documents the boulder size and distribution of a particular coastal boulder bed/ boulder spit. The size of the boulders clearly decreases away from the source area which indicates transport. The form of the boulder ridge also suggests that refraction of the water energy has occurred which has created the curved shape. Figure 6a suggests that the boulders are imbrecated. This pattern should be noted and it could be used to infer the direction of boulder movement. Are the boulders imbrecated along the ridge or across (sea to landward) it.
It is good that a variety of pocesses of formation are discussed but it is noteworthy that a tsunami origin is not considered. The paper outlines the size of storm waves waves required to move the boulders and tries to relate this to observe hurricane waves. A smaller tsunami generated surge would be able to transport the boulders and the sustained flow of water could move many of the boulders in a single event.. The shape of the boulders is ideal for tsunami boulder transport. I would recommend that the discussion section considers a tsunami origin for the transport and that reference is made to some of the research by Ted Bryant. Could the lrgest boulders have been moved by one event and then smaller clasts moved by hurricanes and other storm events.
In terms of detailed suggestions
Lines 14-17 Remove from the abstract ... A previous contribution...to .....by Igneous rocks
Line 31 - replace became with was
Line 45-46 - Remove the sentence Little has been documented...
Line 50 Change to ..... is distinct due to its source...
Line 58 Change to ...recorded waves that crashed...
Section 3.2 - Hydraulic Model - Suggest you also consider the equations associated with tsunami waves and, block movement - see Ted Bryant's book on Tsunamis. Also see the paper by Mottershead et al 2015 Earth Surface Process and Landforms, Vol 40, p2093-2111.
Section 5. - Need to consider a tsunami origin for boulder movement.
Author Response
Reader #2 provided a short review (especially as compared to Reader #1), requesting only minor revisions. In particular, 5 recommendations were offered and most but not all have been accommodated.
1) Title change in order to attract attention of a wider audience besides experts on Mexican geology.
We thought hard and long about this request. In the end, we decided to keep the title mostly as is. Our previous study on this same topic published in the JMSE (2018) achieved attention far beyond our wildest hopes. As of now (12 June 2019, after less than 6 months) that paper has tallied 585 views and 468 downloads. It means that 80% of those who look at the paper made the decision to down-load. The title of our 2018 paper is equally specific as to a particular study area in western Mexico. The chief identifier in that paper is “hurricane deposit” and the slight adjustment we have made is to use the same key word in the present paper (changed from “storm deposit.” This appears to be an adequate change to assure strong interest by the readership.
2) A request for 10-15 additional references to be added to the paper’s introduction in order to provide a broader theoretical background regarding storm deposits.
Again, the sentiment is correct, but we feel that the request for as many as 15 additional references (up front) is too much. Instead, we have added a more modest number of references for comparison in the discussion section – which relates to the suggestion of Reader #2 in item 4. In particular, the need to include material regarding tsunamis (raised by Reader #3) made this effort truly necessary. There are differences in personal writing style, and there are those authors who front-load a long list of references in their introductions. My preference is for a lean style for the introduction that clearly lays out the problem and scope of the study. However, taking the Reader’s recommendation to heart, we have added a reference to the seminal paper by Ruban et al. (2019) from the mdpi journal “Geosciences” that lays out an extensive list of papers (mostly from the Quaternary) regarding deposits with “megaclasts.” In the revised version of our paper, we take care to categorize the data from these 58 references with particular attention to specific papers arguing for or against tsunami deposits as opposed to storm deposits. Key papers from this important data set are cited, but otherwise the analysis is restricted to the Ruban et al. (2019) paper.
However, a few other papers not mentioned in the Ruban et al. (2019) study are introduced in the discussion. The main object was to point out that many of the Quaternary “storm-deposit” papers deal with limestone, whereas volcanic rocks like basalt and andesite are not covered in the present literature. As our goal is to expand on the topic of storm deposits formed by igneous rocks, this was an important addition to the paper.
3) A request to refer to nomenclature regarding boulders and other particles in the methods section.
This is very important and we now refer specifically to key original papers by Wentworth (1922) and by Ruban et al. (2019) in this regard.
4) A request for a comparison with any other, similar studies in other parts of the world.
This request circles back to item 2 (above). Firstly, a detailed comparison is made with our previous contribution on this topic that entails a limestone boulder deposit also in Baja California, Mexico. Details on tonnage and size of limestone blocks is provided. As stated (above) a selected few articles dealing with tsunamic vs. storm deposits are now added to the paper. From these, we extract specific comparisons regarding comparative details on tonnage and size of “megaclasts.”
5) Tasks for further research should be specified briefly in the conclusion section.
Done. In particular, the last of 5 conclusions is now expanded in order to draw attention to the importance of considering parting planes and vertical joints in volcanic rocks like basalt and andesite that form extensive rocky shores all around the Gulf of California. Essentially, these factors mimic the same aspects of limestone covered in our previous (2018) study.
In summary, the recommendations made by Reader #2 are short, but very direct (compared to Reader #1). We believe that all recommendations have been addressed, at least in spirit.
As the reader has made his identity known to us, he is thanked in the acknowledgements.
Reviewer 3 Report
I have read this article with attention and really enjoyed. This is example of well-thought, informative, and internationally-important work on an issue that is urgent, but rarely referred in the literature. I strongly recommend to accept this paper after MINOR revision. My recommendations are given below.
1) Please, make the title more appealing (it should attract the attention of the international research community, not only experts in SW Mexican geology).
2) Introduction should provide a broader theoretical background of storm deposits studies with more citations to fresh literature (please, add 10-15 sources, at least).
3) Nomenclatures of boulders and other particles (Udden-Wenthworth, Blair & McPherson, Bruno & Ruban, etc.) should be referred in the methodological section of this paper (if even these are not used).
4) In Discussion, I would prefer seeing a comparison to results of any other, similar studies in the other parts of the world.
5) Tasks for further research should be specified briefly in Conclusions.
Author Response
Reader #3 brings up an issue not considered by Readers #1 and #2: the relevance of tsunamis in the formation of coastal boulder beds. This is an important consideration which we have taken great care to address.
The key statement in the Review Report is as follows: “I would recommend that the discussion section consider a tsumani origin for transport and that reference is made to some of the research by Ted Bryant. Could the largest boulders have been moved by one event and then smaller clasts moved by hurricanes and other storm events?”
The most efficient way to respond to this recommendation is to simply provide the text for the completely new section (5.3) added to the paper on this account. This should speak for itself and explain why we reject the hypothesis of tsunami action in the Gulf of California (generally) and in our field area (specifically).
5.3. Extra-regional Tsunami Activity
Deposits correlated with extreme wave action in other parts of the Mexico’s Pacific coastline have raised the question as to whether a source from tsunamis can be distinguished from major storms [17]. In particular, the adjoining Jalisco and Michoacán states of Mexico far south of Baja California Sur are bordered by an active subduction zone resulting from compression between the Rivera lithospheric plate and the continental mainland. Among the historical events recorded for this region, the June 22, 1932 earthquake (magnitude 7.7) was one of the region’s most destructive affecting an area 1 km inland along a 20-km stretch of coast with a run-up of 15 m [18]. In contrast, the Gulf of California has no historical record of tsunami events, although shallow earthquakes are relatively common due to transtensional tectonics associated with activity along multiple strike-slip faults that dissect narrow sea-floor spreading zones [5]. Traces of former subduction zones related to the San Benito and Tosco-Abreojos faults extend offshore along much of the outer Pacific coast of Baja California, but these ceased to be active approximately 12 million years ago [5]. Rocky shores along the inner gulf coast of the Baja California peninsula close to the study site entail steep cliffs that rise abruptly to elevations as high as 100 m (Figure 1c). Nothing has been described as remotely similar to the Pleistocene tsunami deposits documented with a run-up of 270 m against the steep volcanic shores of Santiago in the Cape Verde Islands [19]. The hypothesis of a tsunami origin for the Ensenada Almeja CBB is easily eliminated on account of the barren zone lacking boulders inside the partial-ring construction (Figure 5), as well as the occurrence of the CBB restricted to one side of Ensenada Almeja. Any potential tsumani source would have filled the interior of the half-ring with boulders. Moreover, comparable deposits can be expected to have formed along the opposing shores of the bay.
The references cited, therein, serve their purpose and therefore we have not looked into the research by Ted Bryant or the paper by Mottershead et al. (2015).
Reader #3 also took care to offer suggestions keyed to specific lines in the original draft. All have been addressed, adopting those specific recommendations.
This is a very important topic (also considered in reference to the paper on “megaclasts” by Ruban et. al. (2009) and an acknowledgment is added to the revised paper to this effect.
Round 2
Reviewer 1 Report
I revised the manuscript and I guess Authors greatly improved it. I have just a few further suggestions.
Regarding the “Introduction” (it is just an observation, final decision is up to Authors): it seems a bit strange to me that the manuscript starts with the description of Hurricane Odile I guess would be more appropriate to start with the content presented at lines 65-67.
As observed in my first revision, the study area (“2. Geographical and Geological Setting”) is very well described by a geological point of view but there is almost nothing about wave climate…I mean tide (periodicity and range), wind (wind rose of approaching directions or almost more indications about velocities and directions…) and waves (height, approaching directions, extreme values, etc.) …this is very important to characterize the area and than for discussion (that was indeed improved in this second version – but a link with the study area characteristics is required).
So I asked around and colleagues told me that marine climate data for the area investigated could be found at NOAA web page for free, i.e. re-analysis wave data ERA 5 or WWIII. There are also data collected since the 80s by CFE, UABC and the CICESE.
Along the text I found a couple of minor errors.
At Figure 1 captions you mention a square to indicate the study area but I was not able to see it in the figure…
At line 441 I suggest to re-phrase and say: “coastal depositional settings”.
Author Response
Response to Review Report #1 (Round 2)
Manuscript ID: jmse-521890
Geomorphology of a Holocene storm deposit eroded from rhyolite sea cliffs on Ensenada Almeja (Baja California Sur, Mexico)
From the second report by Reader #1, minor adjustments are recommended. It is stated that the discussion section related to aspects about the “wave climate” in the Gulf of California was “indeed improved in the second version.” However, the reader has request yet additional information pertaining to tide periodicity – with an explicit request for a “wind rose of approaching directions.”
As to the request for a wind rose, we have cited a paper in our “Atlas” volume on the Gulf of California (Johnson and Ledesma-Vázquez 2009, Univ. of Arizona Press), wherein just such a wind rose is presented. It is based on the orientation of structures in 84 coastal sand dunes throughout the entire region. We feel it is counter-productive to research an entirely new rose diagram based on data from another source, when we possess those data ourselves. To reprint the diagram from a published source would have required us to obtain a copy-right allowance from the U. of Arizona Press. An already published source should be adequate for the purposes of the present contribution.
To back up our own published observations on wind speed during wind events in the Gulf of California, we have added another citation to the work of Merrifield and Winant (1989) with published wind diagrams.
In addition, we added a reference to the work of Avila-Serrano et al. (2009) contributed to the aforementioned “Atlas” volume, in which the extreme tides characteristic of the Upper Gulf of California are described and quoted. We consider it outside the scope of this paper to make any longer diversion regarding tides not at all characteristic of our study site in the mid-latitude region of the Gulf of California.
Otherwise, Reader #1 found a couple minor errors. Both of which have been attended to. The most important was the observation that the revised map in Figure 1a showed a square for the study site so small it was difficult to see. Thus, for the second time, we have revised Figure 1, in this case to insert a much larger square. The larger box represents not only the study site, but also the entire region shown in the adjoining Figure 1b. We trust this is adequate.
Finally, Reader #1 expressed the opinion that it was somewhat strange to start of the first paragraph of the Introduction section with a statement about Hurricane Odile from 2014. But, whether or not to relocate this passage later in the text was left up to the authors.
We prefer to keep the introductory sentences just as they are, because we wish to make a strong statement from the “get-go” that hurricanes do (in fact) affect the Gulf of California. And in particular the storm waves from Hurricane Odile were filmed at a rocky-shore locality not more than one kilometer from our study site. The video is attached to our contribution under Supplementary Materials.
Thus, with this response in hand that includes the citation of two additional sources and the second revision of Figure 1, we hope a final adjudication of our contribution may now be made. All co-authors were advised of the latest changes to the manuscript and agree.
